# Succession of Microbial Community during the Co-Composting of Food Waste Digestate and Garden Waste

**DOI:** 10.3390/ijerph19169945

**Published:** 2022-08-12

**Authors:** Xiaohan Wang, Xiaoli He, Jing Liang

**Affiliations:** 1Shanghai Academy of Landscape Architecture Science and Planning, Shanghai 200232, China; 2Shanghai Engineering Research Center of Landscaping on Challenging Urban Sites, Shanghai 200232, China

**Keywords:** co-composting, food waste digestate, garden waste, microbial community

## Abstract

Microorganisms are of critical importance during the composting process. The aim of this study was to reveal the bacterial and fungal compositions of a composting pile of food waste digestate and garden waste, where the succession of the microbial communities was monitored using Illumina MiSeq sequencing. We explored the efficiency of composting of different microorganisms to judge whether the composting system was running successfully. The results showed that the composting process significantly changed the bacterial and fungal structure. Firmicutes, Proteobacteria, and Bacteroidota were the dominant phyla of the bacterial communities, while Ascomycota was the dominant phylum of the fungal communities. Moreover, the highest bacterial and fungal biodiversity occurred in the thermophilic stage. The physical and chemical properties of the final compost products conformed to the national standards of fertilizers. The efficient composting functional microbes, including *Cladosporium*, *Bacillus* and *Saccharomonospora*, emerged to be an important sign of a successfully operating composting system.

## 1. Introduction

Food waste (FW) refers to food that is considered suitable for consumption but wasted or lost at the retail or consumption phases, which is generally generated from restaurant, vegetable market, household kitchen, etc. [1,2]. With the rapid development and change of global urbanization and people’s lifestyles, food waste from various industrial, agricultural, and household sources has increased dramatically [3]. The United Nations Environment Program (UNEP) reported that 931 Mt of food was wasted globally in 2019 [4], with 61% of this waste resulting from households and 39% from food service and retail. In the European Union, around 88 million tons of FW are generated annually [5]. In the United States, about 40 percent of edible food is not consumed, resulting in an estimated 37 Mt of food being wasted every year [6]. An estimated 56.57 Mt of FW was generated in 2015 in mainland China [7]. Meanwhile, the continuing population and global consumption growth increased the vulnerability of food systems, disrupting regional supply chains and exacerbating the FW problem.

FW has led to negative outcomes in terms of environmental, social, and economic sustainability in recent years. FW consumption and disposal has become a widely discussed global issue in the 21st century [8]. Based on an analysis conducted by the Intergovernmental Panel on Climate Change (IPCC), about 8 and 10% of the greenhouse gas emissions (GHG) responsible for global warming were caused by food loss and waste between 2010 and 2016 [9]. Yearly, about 40–50% of food produced in America is wasted, resulting in a cost of about USD 750 billion. Most of this waste goes to landfills, accounting for 21% of America’s landfill volume and 9% of the total GHG [10]. The amount of kitchen waste produced in China increased by 100% over the next five years to reach 125 million tons in 2020 [11]. It was estimated that 30.1 Mt oil-Eq of fossil fuels and 16.7 Mt of fresh water were consumed and 37.5 Mt of CO_2_-Eq. M was released into the atmosphere due to this [11]. Nevertheless, approximately 90% of this waste still currently enters incinerators and landfills, to be combined with other municipal solid wastes. Many of the communities across the country are working towards achieving zero waste. In May 2019, the Ministry of Ecology and Environment of the People’s Republic of China carried out the action towards becoming a “Zero Waste City”, making an effort to promote waste reduction, improve the resource utilization of solid waste, minimize the amount of FW entering landfills, and reduce its impact on the environment. Resource recycling has become the main development trend of FW in China [2].

Anaerobic composting is an environmentally friendly treatment option for FW and other organic waste. FW contains a high proportion of easily degradable organic matter, resulting in its rapid degradation and higher production of methane [12]. However, the C/N ratio of China’s FW is generally less than 20 and it lacks trace elements. Co-composting can regulate the physical and chemical properties of the substrate and has been widely used to improve compost performance [13]. Garden waste (GW) is also commonly produced in urban management activities. GW is an important renewable biomass resource with a high lignocellulose content, leading to it having difficulties in degrading, a longer digestion time, and lower methane emissions during the digestion process [14]. By mixing garden waste with food waste, the C/N ratio can be balanced while avoiding the accumulation of volatile fatty acids during the rapid decomposition of FW. In addition, the co-composting of GW and FW achieves the synergistic management of both of the types of waste, which account for a high proportion of municipal solid waste in many developing countries [15,16].

As a biological process, the activity of aerobic microorganisms is the key factor driving the composting process. It was hypothesized that the composting feedstock greatly influenced the microbial community and its evolution in each stage of composting [17,18]. High-throughput sequencing technology has been widely used to investigate the microbial mechanisms of the compost maturation process of various feedstocks. Despite growing attention in the co-composting of FW digestate and other organic materials, such as sewage sludge [19] and biochar [20], the mechanism of the FW digestate co-composting with GW has received limited focus. In addition, how the GW as a co-composting substrate affects the maturity and biological mechanism during FW composting is still poorly understood. The aim of this study was therefore to investigate the changes that take place in the microbial communities in a composting experiment with a municipal FW digestate and GW as the input substrates. Our aim was to determine how the microbial composition would evolve during the composting process.

## 2. Materials and Methods

### 2.1. Material Source and Composting Operation

Fresh GW and FW digestate were used as the input substrates for this experiment (Table 1). The GW was collected from a disposal point in Shanghai, China. The GW was produced from wood chips, yard trimmings, and tree cuttings and then shredded to an appropriate size before the composting trials. The FW digestate was collected from a FW anaerobic digestion factory in Shanghai and was produced by an industrial mesophilic anaerobic digestion process. The basic physicochemical properties of the FW digestate and GW are presented in Table 1.

An open windrow composting experiment was conducted, using a mixture of FW digestate and GW. The GW was mixed with FW digestate to adjust the moisture level to 60% and the initial C/N ratio was 22.45 before the composting begin. The GW proportion was 30% and it was fully and evenly mixed with the FW digestate. The composting process lasted for 91 d, until the germination index (GI) of the composting samples was higher than 60%. The center temperature of the composting pile was measured every 14 days except for in the first 7 days of the process. The initial pile top area and height were 1 m^2^ and 2 m, respectively. The ventilation frequency was 2.00 L min kg^−1^·VS (volatile solid). The composting samples were collected on days 0, 7, 21, 35, 49, 63, 77, and 91 to determine their physicochemical properties, and microbial analyses were conducted during the composting process.

### 2.2. Analytical Methods

The composting samples were dissolved in ultrapure water in a ratio of 1:10 (weight: volume), and the extracted solution was filtered using a 0.45 μm filter membrane to measure the pH, conductivity (EC), ammonium nitrogen (NH_4_^+^-N), and GI. The pH and EC were measured using a pH/EC meter (OHAUS Starter 2100 and ST3100C, Pine Brook, NJ, USA). The NH_4_^+^-N contents were measured using a segmented flow analyzer (Technicon Autoanalyser System, AA3, SEAL, Germany). The GI of the composting sample was determined in three replicates, according to previous studies [1]. The extracted solution was added to a Petri dish with a filter paper sheet placed at the bottom, and 20 Xiaobaicai seeds were evenly placed on the filter paper. Next, the Petri dish was incubated in the dark at 25 °C for 72 h. For this experiment, 5 mL of ultrapure water was used as the control. The following Equation was used to calculate the GI of each sample:$$\mathrm{GI}=\frac{Seed\;germination\;number\;of\;sample\times Root\;length\;of\;sample}{Seed\;germination\;number\;of\;control\times Root\;length\;of\;control}\times 100$$

The remaining fresh samples were air-dried and then ground to a powder to analyze their contents of total organic matter (TOM), total carbon (TC), and total nitrogen [21] following the standard for organic fertilizer (NY/T 525-2021) [22].

### 2.3. Microbial Community Analysis

#### 2.3.1. Sampling, DNA Extraction and High-Throughput Sequencing

All of the composting samples were pre-treated, according to a method described in a previous study [23]. The total genomic DNA of the composting samples was extracted from the prepared samples, using the soil DNA kit (Omega, Bio-Tek, Norcross, GA, USA) according to the manufacturer’s instructions. The DNA extracted for each compost sample was then pooled to gain a representative sample and stored at −20 °C for further analysis. After that, the PCR amplification of the V3–V5 hypervariable regions of the bacterial 16S rDNA genes was performed with the bacterial primers 341F (CCTACGGGAGGCAGCAG) and 907R (CCGTCAATTCCTTTRAGTTT) [24]. Additionally, the ITS1 and ITS2 regions of the fungi were amplified using the primer sets “GTGAATCATCGARTC” and “TCCTCCGCTTATTGAT” [1]. The total PCR products were purified by AMPureXp beads (Beckman Coulter, Inc., Pasadena, CA, USA) and quantified using the Illumina MiSeq PE300 platform (Illumina, Inc., San Diego, CA, USA) to analyze the bacterial and fungal communities in each sample. In this study, the result of the 16S rDNA and ITS rDNA sequences per sample exceeded 69,000, which was sufficient to identify the full microbial diversity of the co-composting process.

#### 2.3.2. Bioinformatics and Statistical Analyses

The sequence data processing was conducted on QIIME Pipeline-Version 1.9.0 (https://qiime.org/ (accessed on 1 March 2021)). All of the sequence reads were trimmed and assigned to each, according to their bar code. The sequences with a high quality (length > 300 bp, without ambiguous base ‘N’ and average base quality score > 30) were used for downstream analysis. The sequence clustering involves an operation classification unit (OTUs) with a consistency threshold of 97%. The taxonomy was performed, using the Ribosomal Database Project Classifier-Version 2.2. The R software (https://www.r-project.org/ (accessed on 1 March 2021)) was used for subsequent statistical analyses.

## 3. Results and Discussion

### 3.1. Evolution of Physicochemical Properties

The physicochemical indicators, including SOC, pH, temperature, C/N ratio, and GI, are commonly used to characterize the aerobic composting process. The main physicochemical properties of food waste with greening litter during the composting process are summarized in Table 2. The maximum temperature was only 68.2 °C and the high temperature stage persisted for the first 7 days. The temperature changes during the composting process indicate the decomposition of the organic matter by microorganisms, while to a certain extent, destroying pathogens and other harmful components. The initial increase in the temperature ensured pathogen destruction [25]. A similar temperature profile was also observed during the composting of the FW digestate with sawdust [26]. Compared with the composting of the FW digestate only, the addition of GW may lead to an earlier cooling stage [26]. The pH increased from D7 and reached the peak (9.2) at D21. The increase in the pH was mainly due to the release of ammonia, leading to ammonification [27]. Afterwards, the pH remained stable at 8.7–8.9. The pH stabilization near the process completion could be attributed to the decline in ammonification, due to the reduction in the NH_4_^+^-N content [26]. The EC value reflects the soluble salt content of the compost, which, to a certain extent, indicates the inhibition and toxicity effect on plant growth after application in the soil. During this process, the EC increased from 3.4 to 4.2 mS·cm^−1^. The growth of the EC probably occurred due to the relatively high concentration of soluble salts from the raw material and simultaneous moisture loss. The sample final EC met the Hong Kong [28] and Australian [29] standards for composts. Moisture affects the microbial community activity and the biochemical processes it drives. The samples’ moisture contents varied between 24.8% and 42.8%. The moisture loss that occurred during composting could be considered as an indicator of the rate of decomposition, as the heat generated during decomposition drove vaporization [30]. The OM is also a critical index of composting maturity, with the easily degradable compounds mineralizing into carbon and being utilized as nitrogen sources for microbials in the process of aerobic composting [31]. The OM content decreased during the composting process, and this decrease was more pronounced after D35. The OM loss was 21.26%. GI is a widely used indicator of compost maturity, because it is a direct measure of the phytotoxicity of compost products to seed germination and seed growth [32]. The total nitrogen variations in the composting may be associated with the strong ammonia volatilization caused by the ammonia stripping [33]. As the composting process proceeded, the GI value increased sharply from D63 and finally reached a maximum of 62.2% on D91. As a key parameter to evaluate the maturity of the compost, the threshold of GI has been clearly specified in the relevant standards, such as the Compost and Soil Conditioner Quality Standards [28]. Generally, a GI value lower than 50% is an indication of the high phytotoxicity and maturity of the compost. Judging from the evaluation of the GI value, the compost product achieved maturity after composting for more than 77 days. Considering the properties discussed previously, the maturity parameters selected indicated that the co-composting of the FW digestate and GW finally reached maturity.

### 3.2. Microbial Community Profile in Composting Systems

#### 3.2.1. Bacterial Composition and Diversity

To better understand the changes in the microbial communities of the composting treatment, the diversity of the bacterial and fungal communities was examined using the Chao1, Shannon evenness index, and Simpson diversity indices. The Chao1 estimator is one of the most commonly used indices for measuring the richness of environmental microorganisms, and a higher Chao1 value indicates the presence of more species. The Shannon evenness index shows the balance between species. The Simpson index combines evenness and richness to characterize species diversity. Due to the greater weight of dominant species, rare species have less impact on this index.

A total of 402,009 sequences were obtained from the eight samples sequenced. The results showed that the Chao 1 index of the bacteria increased from D0 to D49 but decreased on D63 (Table 3). The bacterial Shannon evenness index increased gradually after the composting began and reached its maximum value on D63, after which the change was small. Furthermore, the bacterial Simpson index increased sharply from D21 to D35 after composting and then decreased rapidly.

A very diverse bacterial community was detected in the samples after the composting began. The relative abundances of the bacterial genera during the composting processes are presented in Figure 1. Three phyla showed the highest number of OTUs—Firmicutes, Proteobacteria, and Bacteroidota—and their relative abundances were 46.80%, 16.69%, and 16.41%, respectively. These results are in accordance with the data presented by Wang et al. [34], who also showed that Proteobacteria and Firmicutes were the dominant phyla in the four stages of sludge composting. Firmicutes can produce different extracellular enzymes (e.g., cellulase, lipase, protease), so the presence of this phylum reflects its ability to metabolize a variety of substrates during the composting process, including proteins, lipids, lignin, cellulose, sugars, and amino acids [21]. Clearly, at the beginning of composting, W5 (phylum *Cloacimonetes*) was the most abundant bacterial genera in the D0 sample, as its relative abundance was 26.51%. After the composting began, the bacterial composition changed significantly. From D7 to D35 of composting, the dominant group of bacterial genera in the samples changed to *Bacillus* (phylum Firmicutes), *Pseudogracili-bacillus* (phylum FirmicutesFirmicutes), and *Oceanobacillus* (phylum Firmicutes). Due to its thermo-tolerance, the genus *Bacillus* is ubiquitous in lignocellulose composting and contributes to waste degradation during this process [35]. Despite minor differences observed in the relative abundances of the D7, D21, and D35 samples, there was no significant difference in their bacterial phylogenetic groups. The relative abundance of *Bacillus* (phylum Firmicutes) and *Pseudogracilibacillus* (phylum Firmicutes) decreased rapidly in the middle and late compost periods (D49 to D77), while the relative abundance of *norank_f_Fodinicurataceae* (phylum Proteobacteria), *Truepera* (phylum Deinococcus-Thermus), and *Actinoadura* (phylum Actinobacteria) increased significantly. At the end of the composting, the bacterial dominant groups on D91 were completely different from those at D0. The dominance of Firmicutes throughout the thermophilic phase could be attributed to its ability to survive under thermophilic conditions [36]. The most abundant bacterial genera in the D91 sample were *norank_f_Fodinicurataceae* (phylum Proteobacteria), *Trueera* (phylum Deinococcus-Thermus), and *unclassified_f_Flavobacteriaceae* (phylum Bacteroidetes), with relative abundances of 21.03%, 10.46%, and 6.39%, respectively.

#### 3.2.2. Fungal Composition and Diversity

Compared with the richness of the bacterial community, the Chao1 of the fungal community decreased significantly with the increase in the composting time and reached the lowest at the end of composting. Similar change trends were identified by the fungal Shannon evenness index.

A total of 606,024 sequences were obtained from the eight samples sequenced. A total of 1212 fungal OTUs were obtained, which were taxonomically assigned to 14 phyla, 43 classes, 87 orders, 181 families, and 322 genera. The most abundant fungal communities belonged to the phylum Ascomycota, with a relative abundance of 76.61%. Ascomycota was a widely distributed fungal phylum in many compost processes because of its ability to rapidly adapt to changing environmental factors during the composting process [37]. In particular, previous studies have shown that it can withstand environments with high temperatures and low moisture contents [38]. Moreover, Ascomycota is one of the most important fungi that is beneficial to the degradation of lignocellulose and other organic matter in the organic composting process [39].

Figure 2 shows the apparent variations in fungal genera during composting. At the beginning of the composting, the most abundant reads belonged to the genus *Kazachstania* (phylum Ascomycota), with a relative abundance of 90.16%. During the thermophilic stage (D7 to D35), the relative abundance of *Cladosporium* (phylum Ascomycota) and *Aspergillus* (phylum Ascomycota) gradually increased, indicating that the most dominant fungi were gradually replaced by thermophiles. Noticeably, at the initial stage of composting (D7), the relative abundance of *Mortierella* (phylum Mucoromycota) and Geastrum (phylum Basidiomycota) increased significantly. The results of Zhang et al. [38] have shown that this fungal genus was involved in the decomposition of fulvic acids and unextractable humin, as well as the formation of humic acids. During the cooling stage, the relative abundance of *Sodiomyces* (phylum Ascomycota) increased rapidly and reached the highest 94.63% on D63. During the final maturation phase (D 77 to D91), *Sodiomyces* (phylum Ascomycete) and *unclassified _p_ Ascomycota* (phylum Ascomycota) became the two dominant genera. The dominance of fungi that are capable of lignocellulose degradation pointed out the efficiency of the process in the compost piles [37]. Other than these commonly found compost fungi, the genera *Hapsidospora* (phylum Ascomycete) and *Phialosimplex* (phylum Ascomycete) were also examined in the final mature compost. Among them, some representative species of the genus *Phialosimplex* were found to exist widely in hypersaline habitats [40].

### 3.3. Potential Mechanisms of the Co-Composting Efficiency

It is well known that the microorganisms are the key drivers in the composting process. Therefore, the characteristics of microbial composition and diversity in the composting ecosystem are very important for the construction and application of an efficient composting system.

The environmental variables have been proposed to be important for microbial community shaping during the composting. A redundancy analysis was applied to determine the correlation between the gene abundances and environmental variables, including pH, EC, temperature, water content, organic matter, total nitrogen, and C/N ratio. In the bacterial RDA plot (Figure 3A), the first two components explained 99.16% of the total variation. For the fungal community, the first two components explained 96.72% of the total variation (Figure 3B). The results indicated that these environmental parameters are major factors shaping the microbial community for both bacteria and fungi. Through the composting process, temperature (*p* = 0.04) was found to statistically explain the bacteria variation. All of the environmental variables failed to statistically explain the fungi variation. The importance of the role of temperature in microbial community dynamics has been widely recognized [41]. As shown in Figure 3A, the bacteria phylum Proteobacteria had positive correlations with pH and EC and existed in the late compost period (D49 to D91), suggesting the phylum were able to exist with high pH and EC. As show in Figure 3B, the phylum Ascomycota had positive correlations with EC and existed in the middle and late compost periods (D49 to D91), suggesting the phylum could grow in a high EC environment. The results indicating that environments suitable for microbial growth form gradually during the composting process [42].

Co-composting feedstock also can change the microbial community structure and diversity, thereby affecting composting maturation [18]. Akyol et al. [37] composted crop straw, such as barley and wheat, and found that *Luteimonas* (phylum Actinobacteria) and *Thermomyces* (phylum Ascomycota) were the most abundant bacterial and fungal genera, respectively. However, neither of these microorganisms were the dominant microorganism during the entire co-composting process in the present study. It has been shown that the thermophilic stage of the composting processes was the main stage of material transformation, compared with the mesophilic stage and the mature stage, which depend on the high enzyme activity of different indigenous thermophilic microorganisms [43]. Then, thermophilic bacteria and fungi play a leading role in increasing the composting efficiency. In the present study, the most abundant bacteria phylum Firmicutes had a positive relation with the temperature and organic matter in the middle compost period (D35) (Figure 3A). The results support that the abundance of Firmicutes was associated with the decomposition of organic matter during the thermophilic stage [44]. The present study also showed that the relative abundance of *Cladosporium* belonging to the phylum Ascomycota increased during the thermophilic stage. A recent study conducted by Wang et al. [1] found that the *Cladosporium* genus first developed in the most closely attached fraction of the microbe to the activated carbon, promoting the degradation of lignocellulose matter. In addition, *Bacillus*, which has been shown to promote organic biodegradation [13], was also increased during the thermophilic stage. From D21 onwards, the genus *Saccharomonospora* belonging to the phylum Actinobacteria emerged to facilitate the transformation of lignin and cellulose into humus [45]. Although the Proteobacteria phylum could exploit small molecules (i.e., propionate and butyrate), it could not withstand the high temperatures [46]. The emergence of efficient composting functional microbes is an important sign of the successful operation of a composting system. The FW digestates can be effectively co-composted with lignocellulose-rich amendments, which can enhance the nitrification and decomposition rates [47]. Song et al. [26] found that mixing FW digestate compost with sawdust or mature compost led to a lower NH_4_^+^-N and higher GI (>80%) within 2 weeks. Manu et al. [20] conducted a laboratory batch-composting study and showed that the addition of 10% biochar could significantly enhance the nitrification and humification process during the co-composting of FW digestate. The present study showed that the system of co-composting FW digestate with GW resulted in a reduction in the total organic matter content from 44.2% to 34.8%. The composting systems were mostly dominated by lignocellulose degraders, ensuring an efficient composting process [37].

## 4. Conclusions

In this study, we investigated the bacterial and fungal composition of a co-composting pile of FW digestate and GW and analyzed the succession of the microbial communities present. We found significant changes in the bacterial and fungal structure throughout the co-composting process. Firmicutes, Proteobacteria, and Bacteroidota were the dominant phyla of the bacterial communities, while Ascomycota was the dominant phylum of the fungal communities. The characteristics of the final products were in line with related maturity parameters. The functional microbes, including *Cladosporium*, *Bacillus*, and *Saccharomonospora*, may be used as important signs with achieving a more efficient co-composting system.

## Figures and Tables

**Figure 1 ijerph-19-09945-f001:**
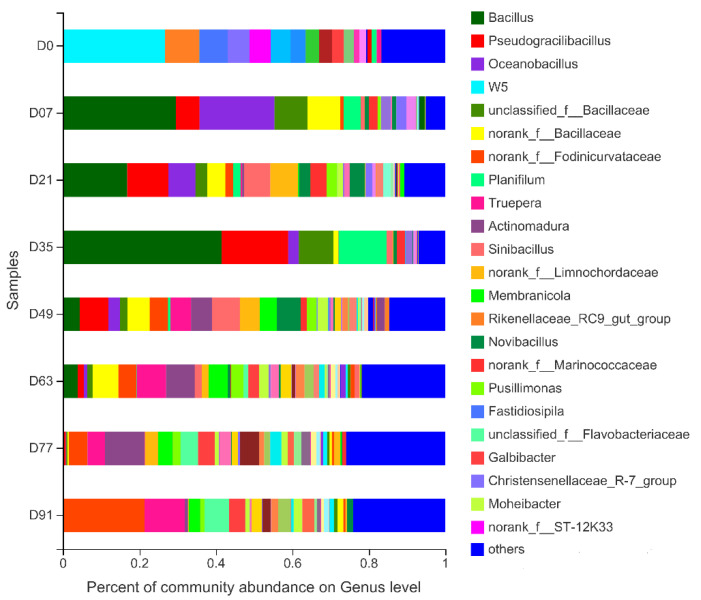
Genus levels composition of the bacterial community during the composting process.

**Figure 2 ijerph-19-09945-f002:**
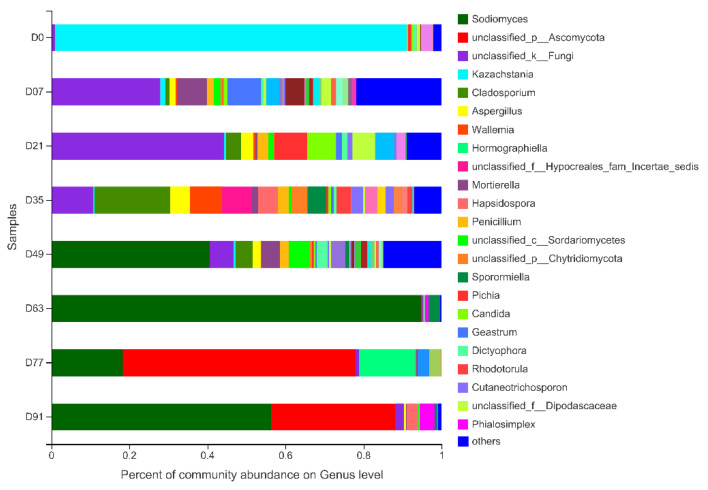
Genus levels composition of the fungal community during the composting process.

**Figure 3 ijerph-19-09945-f003:**
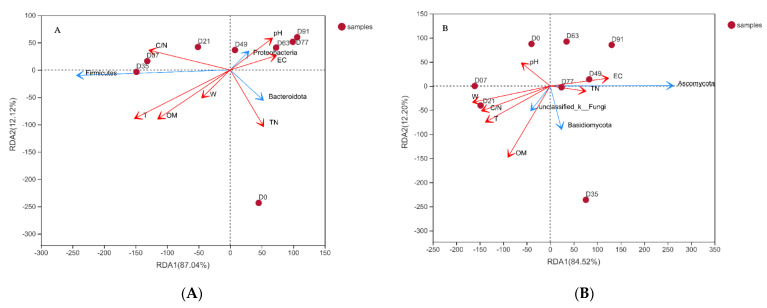
Redundancy analysis (RDA) of the bacterial (**A**) and fungal (**B**) communities and the environmental variables (W): water content; OM: organic matter; TN: total nitrogen; T: temperature). The environmental variables represented in the RDA are shown as red vectors in the plots. The dominant phyla of the bacterial and fungal communities are shown as blue vectors.

**Table 1 ijerph-19-09945-t001:** The basic physicochemical characteristics of GW and FW digestate (TN and TOM correspond to total nitrogen and total organic matter).

Materials	PH	EC (mS cm^−1^)	TN (%)	C/N	TOM (%)
GW	7.69	0.83	1.63	35.14	98.80
FW digestate	9.13	3.11	3.57	7.20	44.33

**Table 2 ijerph-19-09945-t002:** Main physicochemical properties of compost samples during composting process.

Parameters	D0	D7	D21	D35	D49	D63	D77	D91
Temperature (°C)	60.8	68.2	54	62.5	45.6	56.2	47.1	43.2
pH	8.8	8.7	9.2	8.7	8.8	8.9	8.9	8.9
Moisture (%)	39.5	39.0	42.8	33.5	32.7	32.7	43.5	24.8
EC (mS·cm^−1^)	3.4	3.5	2.7	3.6	3.4	3.1	4	4.2
TOM (%)	44.2	38.7	46.0	46.4	37.5	35.1	40.2	34.8
C/N	8.27	15.91	9.85	10.19	7.88	9.04	10.23	8.43
TN (g·kg^−1^-TS)	31.0	14.1	27.1	26.4	27.6	22.5	22.8	23.9
GI (%)	0	14.2	7.0	10.6	10.9	20.6	56.0	62.2

**Table 3 ijerph-19-09945-t003:** The bacteria/fungi diversity of samples during the composting process.

Sampling ID	Richness (Chao1) Bacteria/Fungi	Evenness (Shannon Evenness) Bacteria/Fungi	Shannon Bacteria/Fungi	Simpson Index Bacteria/Fungi
D0	463.25/184.78	0.58/0.16	3.43/0.85	0.09/0.74
D7	493.42/209	0.52/0.69	3.10/3.68	0.09/0.09
D21	612.86/260	0.60/0.63	3.72/2.57	0.05/0.21
D35	561.21/267	0.45/0.63	2.67/3.38	0.13/0.06
D49	614.55/217	0.65/0.59	4.12/3.18	0.03/0.18
D63	584.03/86.80	0.73/0.37	4.51/1.32	0.02/0.90
D77	597.53/38.43	0.72/0.33	4.46/1.21	0.03/0.41
D91	525.60/36.00	0.66/0.33	4.08/1.19	0.04/0.42

## Data Availability

Data available on request.

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
