# Peer review of "Succession of Microbial Community during the Co-Composting of Food Waste Digestate and Garden Waste"

_ijerph, 2022, doi:10.3390/ijerph19169945_

Round 1

Reviewer 1 Report

The paper was to reveal the bacterial and fungal composition of the composting pile of food waste digested and garden waste and the succession of the microbial communities were monitored by Illumina MiSeq sequencing. The following comments may be useful to improve the paper.

1. Explain the advantages of co-composting compared with separate composting, and get the answer of why co-composting of these two kinds of garbage should be adopted.

2. The aim of the study mentioned in the abstract only stays at the stage of experimental purpose, which can further clarify its practical significance, and can be combined with the significance such as "finding out efficient composting microorganisms to judge whether the composting system is running successfully"

3. The introduction is biased. It introduces food waste in a large space to highlight the necessity of food waste treatment, and ignores the related introduction of garden waste. It may be more pertinent to add in the article.

4. Small details: The first abbreviation in line 37 is FW instead of WF according to the reader's inference, and WF appears abruptly.

5. Whether the samples of food waste and garden waste in the materials and methods section are representative or not has not been explained.

6. Due to the lack of expansibility, there is a prospect of further research on how composting changes the structure of bacteria and fungi, its causes and significance, and systematic research.

Reviewer 2 Report

The manuscript employed the Illumina MiSeq PE300 sequencing approach to analyze the composition of bacteria and fungi, as well as the succession of the indigenous microbial communities during the co-composting of food waste digested and garden waste. It was illustrated successful operating composting system by physicochemical properties and efficient composting functional microbes. This study contributes to a better understanding of composting driven by microbes, which is a valuable resource for degradation of different types of organic matter.

However, I have some concerns about the incomplete and/or imprecise description of the methodology and results, which hinders accurate understanding of the work. Moreover, the writing of the manuscripts still needs to be further polished. Most importantly, the innovations of experimental settings need to be further clarified. Here, in my opinion the manuscript is not suitable for publication in the present form and require a careful revision as indicated below.

Comments:
1. Line 37: “WF”? Unify the use of abbreviations/acronyms, such as food waste/FW...

2. Line 58: “digest” into “digested”

3. Why not about the content of garden waste in introduction, such as waste generation or current treatment and so on?

4. Line 72: check the data “35.14”, which is the C/N of garden waste, not the mixture before the composting begin.

5. Provide more details about the composting pile, including scale, ventilation and more.

6. Why not the duplications of an experimental treatment? How to ensure the reliability and representative of data? It was not found in both Methods and Results sections.

7. Table 1: define “TN” and “TOM”, the same below, revise “mS·cm-1”.

8. Line 108: Provide average number of Illumina Miseq PE300 platform reads per sample. Are the number of reads sufficient to assess the full microbial diversity in these composting processes?

9. Line 132: “()”? Add relevant information.

10. Line 134: The EC value reflects the soluble salt content of the compost.

11. Line 161: Elaborate the co-composting in study how meet with the all maturity parameters reported.

12. Line 181: The writing criterion of the phylum, capitalize the first letter, no italics, the same below.

13. Fig.1: just display the key genus in figure. Please comply with consistent expression “Day/D”, and Genus name in italics. The Figure 2 is the same problem.

14. In section 3.3: Clarify the logical relationship among feedstock, physiochemical index and function microbial evolution. Assessment of microbial evolution is lack of direct experimental basis and control treatment.

15. Line 273: the important conclusion “The emergence of efficient composting functional microbes…” is lack of direct experimental data support.

16. Line 276: verify the title number.

17. References: keep journal abbreviations consistent.

Round 2
